# A high-throughput neutralizing antibody assay for COVID-19 diagnosis and vaccine evaluation

Antonio E. Muruato[1,2,10], Camila R. Fontes-Garfias[1,10], Ping Ren[3,10], Mariano A. Garcia-Blanco [1,4,5], Vineet D. Menachery[2,3,6], Xuping Xie [1✉] & Pei-Yong Shi [1,6,7,8,9✉]

Virus neutralization remains the gold standard for determining antibody efficacy. Therefore, a high-throughput assay to measure SARS-CoV-2 neutralizing antibodies is urgently needed for COVID-19 serodiagnosis, convalescent plasma therapy, and vaccine development. Here, we report on a fluorescence-based SARS-CoV-2 neutralization assay that detects SARS-CoV-2 neutralizing antibodies in COVID-19 patient specimens and yields comparable results to plaque reduction neutralizing assay, the gold standard of serological testing. The fluorescence-based neutralization assay is specific to measure COVID-19 neutralizing antibodies without cross reacting with patient specimens with other viral, bacterial, or parasitic infections. Collectively, our approach offers a rapid platform that can be scaled to screen people for antibody protection from COVID-19, a key parameter necessary to safely reopen local communities.

[1] Department of Biochemistry and Molecular Biology, University of Texas Medical Branch, Galveston, TX, USA. [2] Department of Microbiology and Immunology, University of Texas Medical Branch, Galveston, TX, USA. [3] Department of Pathology, University of Texas Medical Branch, Galveston, TX, USA. [4] Programme in Emerging Infectious Diseases, Duke-NUS Medical School, Singapore, Singapore. [5] Department of Internal Medicine, University of Texas Medical Branch, Galveston, TX, USA. [6] Institute for Human Infections and Immunity, University of Texas Medical Branch, Galveston, TX, USA. [7] Sealy Institute for Vaccine Sciences, University of Texas Medical Branch, Galveston, TX, USA. [8] Sealy Center for Structural Biology & Molecular Biophysics, University of Texas Medical Branch, Galveston, TX, USA. [9] Institute for Translational Science, University of Texas Medical Branch, Galveston, TX, USA. [10]These authors contributed equally: Antonio E. Muruato, Camila R. Fontes-Garfias, Ping Ren. ✉email: xuxie@UTMB.edu; peshi@UTMB.edu

The ongoing coronavirus disease 2019 (COVID-19) pandemic is caused by severe acute respiratory syndrome coronavirus 2 (SARS-CoV-2), first reported in Wuhan, China in late 2019[1,2]. As of June 28, 2020, COVID-19 has caused 10.3 million confirmed infections and over 505,741 deaths worldwide (https://www.worldometers.info/coronavirus/). Many areas of the world have been in lockdown mode to curb the viral transmission, but the reality is that COVID-19 is here to stay until a safe and efficacious vaccine becomes available. The pandemic's catastrophic economic impact is pushing governments to reopen their economies, and this creates a public health quandary. At this time, our only option is to minimize viral transmission through social distancing and contact tracing, which relies on the diagnosis of viral RNA through reverse transcription polymerase chain reaction (RT-PCR) (https://www.fda.gov/media/134922/download). Proper public health policy would be greatly enhanced if we had a reliable and facile assay to measure the immune protection among COVID-19 recovered patients.

Coronavirus infections typically induce neutralizing antibody responses[3]. The seroconversion rates in COVID-19 patients are 50% and 100% on day 7 and 14 post symptom onset, respectively[4]. Given the unknown scale of asymptomatic infections, there is a pressing need for serological diagnosis to determine the real number of infections. Such information is essential for defining the case-fatality rate and for making the policy on the scale and duration of social lockdowns. The serological assays are also required to identify donors with high-neutralizing titers for convalescent plasma for therapy, and to define correlates of protection from SARS-CoV-2. While viral RNA-based testing for acute infection is the current standard, surveying antibody protection is a necessary part of any return to social normality.

For serodiagnosis, several COVID-19 assay platforms have achieved FDA emergency use authorizations (EUA), including ELISA[5] (https://www.fda.gov/media/137029/download), lateral flow immunoassay (https://www.fda.gov/media/136625/download), and Microsphere Immunoassay (https://www.fda.gov/media/137541/download). These assays measure antibody binding to SARS-CoV-2 spike protein. Since not all spike-binding antibodies can block viral infection, these platforms do not functionally measure antibody inhibition of SARS-CoV-2 infection. An ideal serological assay should measure neutralizing antibody levels, which should predict protection from reinfection. Conventionally, neutralizing antibodies are measured by plaque reduction neutralization test (PRNT). Although PRNT and ELISA results generally corelate with each other, particularly when the receptor-binding domain of SARS-CoV-2 spike protein is used as an ELISA antigen (https://www.genscript.com/cpass-sars-cov-2-neutralization-antibody-detection-Kit.html)[6,7], PRNT remains the gold standard for serological testing and determining immune protection[8,9]. However, due to its low throughput, PRNT is not practical for large scale serodiagnosis and vaccine evaluation. This is a major gap for COVID-19 surveillance and vaccine development.

Here, we report a fluorescence-based high-throughput neutralization assay that detects SARS-CoV-2 neutralizing antibodies in patient specimens and yields equivalent results to the gold standard plaque reduction neutralizing assay.

## Results

### A high-throughput fluorescence-based neutralization assay. To fill in the gap for COVID-19 serodiagnosis and vaccine evaluation, we developed a fluorescence-based assay that rapidly and reliably measures neutralization of a reporter SARS-CoV-2 by antibodies from patient specimens. The assay was built on a stable mNeonGreen (mNG) SARS-CoV-2 where the mNG gene was engineered at the ORF7 of the viral genome (Fig. 1a)[10]. The complete sequence of mNG SARS-CoV-2 is described in Supplementary Fig. 1. Figure 1b depicts the flowchart of the reporter neutralization assay in a 96-well format. The assay protocol is detailed in Supplementary Methods. Briefly, patient sera were serially diluted and incubated with the reporter virus. After incubation at 37°C for 1 h, Vero CC-81 cells (pre-seeded in a 96-well plate) were infected with the virus/serum mixtures at a multiplicity of infection (MOI) of 0.5. At 16 h post-infection, the mNG-positive cells were quantitated using a high-content imaging reader (Fig. 1b). It should be noted that Vero CC-81 cells, not Vero E6 cells, were chosen for the mNG assay to enable accurate quantification of fluorescent cells. Sixty COVID-19 serum specimens from RT-PCR-confirmed patients and 60 non-COVID-19 serum samples (archived before COVID-19 emergence) were analyzed using the reporter virus. For some COVID-19-positive specimens, the sample collection days post viral RT-PCR positive were available and are indicated in Table 1. After reporter viral infection, the cells turned green in the absence of serum (Fig. 1c, bottom panel); in contrast, incubation of the reporter virus with COVID-19 patient serum decreased the number of fluorescent cells (top panel). A dose–response curve was obtained between the number of fluorescent cells and the fold of serum dilution (Fig. 1d and Supplementary Fig. 2), which allowed for determination of the dilution fold that neutralized 50% of fluorescent cells ($NT_{50}$). The reporter assay rapidly diagnosed 120 specimens within 24 h: all 60 COVID-19 sera (specimens 1–60) showed positive $NT_{50}$ of 35 to 5711, and all 60 non-COVID-19 sera (specimens 61–120) showed negative $NT_{50}$ of <20 (Table 1).

### Assay validation by plaque reduction test. To validate the reporter virus neutralization results, we performed the conventional PRNT on the same set of patient specimens. All 60 negative sera (specimens 61–120) exhibited $PRNT_{50}$ of <20 (Table 1). Among the 60 positive specimens, 57 sera (specimens 4–60) showed $PRNT_{50}$ of 40 to 3200, whereas 3 sera (specimens 1–3) exhibited $PRNT_{50}$ of <20 (Table 1). The discrepancy between the $PRNT_{50}$ and $NT_{50}$ values for specimens 1–3 is likely due to the early infection time (within 5 days post RT-PCR positive) when neutralizing antibodies just began to develop; this discrepancy suggests that the mNG SARS-CoV-2 assay has a higher sensitivity than the conventional PRNT assay. Nevertheless, a strong correlation was observed between the reporter virus and PRNT results, with a correlation efficiency $R^2$ of 0.85 (Fig. 1e). The results demonstrate that when diagnosing patient specimens, the reporter virus assay delivers neutralization results comparable to the PRNT assay, the gold standard of serological testing.

### Assay specificity. We evaluated the specificity of reporter neutralization assay using potentially cross-reactive sera and interfering substances (Table 2). Two groups of specimens were tested for cross reactivity. Group I included 150 clinical sera from patients with antigens or antibodies against different viruses, bacteria, and parasites. These human specimens were obtained according to two types of diagnostic results: some samples were tested positive for antibodies against specific pathogens (e.g., anti-Chikungunya virus; this group of samples are indicated by prefix "anti" in Table 2); other specimens were collected within 1 to 6 months after the patients were tested positive on pathogen antigens or nucleic acids (e.g., adenovirus antigen; this group of samples are not indicated by prefix in Table 2). Group II consisted of 19 samples with albumin, elevated bilirubin, cholesterol, rheumatoid factor, and autoimmune nuclear antibodies. None of these specimens cross-neutralized mNG SARS-CoV-2 (Table 2), including the four common cold coronaviruses (NL63, 229E,

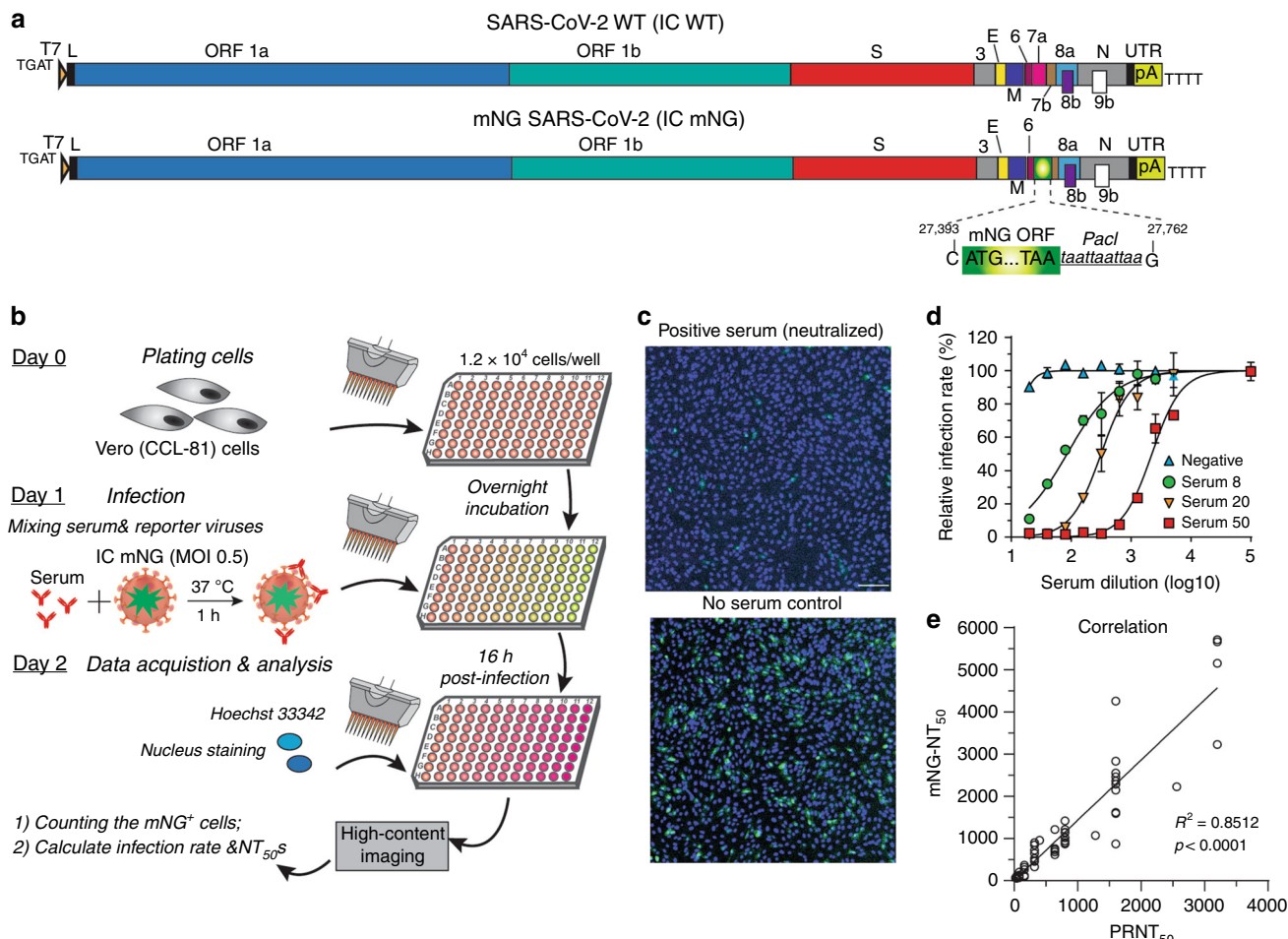

**Fig. 1 A high-throughput neutralizing antibody assay for COVID-19 diagnosis. a** Diagram of the cDNA constructs of wild-type (WT) SARS-CoV-2 (top panel) and mNG SARS-CoV-2 (bottom panel). The nucleotide positions of viral genome where mNG is engineered are indicated. **b** Assay flowchart. mNG SARS-CoV-2 was neutralized with COVID-19 patient sera. Vero CCL-81 cells were infected with the reporter virus/serum mixture with an MOI of 0.5. The fluorescence of infected cells was quantified to estimate the $NT_{50}$ value for each serum. **c** Representative images of reporter virus-infected Vero CCL-81 cells. Images for a positive neutralizing serum (top panel) and no serum control (bottom panel) are presented. Scale bar, 100 μm. **d** Neutralization curves. Representative neutralization curves are presented for three positive sera and one negative sera. The means and standard deviations from two independent experiments are presented. **e** Correlation analysis of $NT_{50}$ values between the reporter virus and PRNT assays. The Pearson correlation efficiency $R^2$ and p-value (two-tailed) are indicated.

OC43, and HUK1). Despite the low number of common cold coronavirus serum specimens, our result is consistent with the recent reports that sera from common cold coronavirus patients did not cross react with SARS-CoV-2[5,11]. More specimens are required to further validate the cross reactivity, particularly between SARS-CoV-2 and other human coronaviruses, including SARS-CoV-1 and MERS-CoV.

### Discussion
In this study, we developed a rapid fluorescence-based high-throughput assay for COVID-19 serodiagnosis. The reporter virus assay is superior to many antigen/antibody binding assays because it measures functional SARS-CoV-2 neutralizing activity in the specimens. When diagnosing patient sera, the reporter virus assay generated $NT_{50}$ values comparable to the conventional PRNT assay. Compared with the PRNT assay, our reporter neutralization test has shortened the assay turnaround time by several days and increased the testing capacity to high throughput. Toward the same direction, VSV and lentivirus pseudotyped with SARS-CoV-2 spike protein have been reported for COVID-19 neutralization assays at biosafety level 2 (BSL-2) lab[12].

Since mNG SARS-CoV-2 is stable and replicates like wild-type virus, our reporter neutralization assay provides an ideal model for high-throughput serological testing. As the mNG SARS-CoV-2 grows to >$10^7$ PFU/ml in cell culture[10], the reporter virus can be easily scaled up for testing large sample volumes. Besides mNG, we have begun to develop other reporter SARS-CoV-2 (e.g., luciferase or mCherry) that can also be used for such serological testing[13]. Although the current study performed the assay in a 96-well format, the assay can be readily adapted to 384- and 1536-well formats. Despite the strengths of high throughput and reliability, the current reporter neutralization assay must be performed in BSL-3 containment. Efforts are ongoing to engineer an attenuated version of SARS-CoV-2 so that the assay could be performed at a BSL-2 facility. Despite the BSL-3 limitation, the mNG reporter assay offers a rapid, high-throughput platform to test COVID-19 patient sera not previously available. Indeed, the mNG SARS-CoV-2 assay is currently being used to support clinical trials for COVID-19 vaccine candidates[14].

As neutralizing titer is a key parameter to predict immunity, the reporter neutralization assay should be useful for high-throughput evaluation of COVID-19 vaccines and for

### Table 1 Comparison of neutralization titers of patient sera analyzed by reporter assay and plaque reduction assay.

| [a]Serum ID[b] | [c]PRNT$_{50}$ | [c]mNG-NT$_{50}$ | [a]Serum ID[b] | [c]PRNT$_{50}$ | [c]mNG-NT$_{50}$ |
|---|---|---|---|---|---|
| 1 (d1) | <20 | 35 | 32 (d9) | 640 | 762 |
| 2 (d5) | <20 | 38 | 33 (d8) | 320 | 846 |
| 3 (d4) | <20 | 50 | 34 (d14) | 800 | 873 |
| 4 (d5) | 40 | 58 | 35 (d16) | 1600 | 874 |
| 5 (d5) | 20 | 66 | 36 (d17) | 320 | 900 |
| 6 (d6) | 80 | 74 | 37 (d9) | 800 | 902 |
| 7 (d8) | 80 | 77 | 38 (d15) | 800 | 949 |
| 8 (d4) | 80 | 85 | 39 (d15) | 400 | 958 |
| 9 (d5) | 80 | 85 | 40 (d18) | 800 | 1016 |
| 10 (d1) | 80 | 95 | 41 (d28) | 1280 | 1072 |
| 11 (d6) | 80 | 96 | 42 (d12) | 800 | 1139 |
| 12 (NA) | 160 | 96 | 43 (d13) | 800 | 1145 |
| 13 (d6) | 40 | 111 | 44 (d14) | 800 | 1210 |
| 14 (d6) | 40 | 114 | 45 (d31) | 640 | 1213 |
| 15 (d1) | 80 | 115 | 46 (d8) | 800 | 1419 |
| 16 (d9) | 160 | 120 | 47 (d14) | 1600 | 1590 |
| 17 (d11) | 80 | 132 | 48 (d21) | 1600 | 1617 |
| 18 (d8) | 80 | 200 | 49 (d12) | 1600 | 2148 |
| 19 (NA) | 160 | 261 | 50 (NA) | 2560 | 2225 |
| 20 (d5) | 160 | 318 | 51 (d20) | 1600 | 2287 |
| 21 (d32) | 320 | 329 | 52 (d8) | 1600 | 2362 |
| 22 (d14) | 160 | 365 | 53 (d12) | 1600 | 2463 |
| 23 (d12) | 160 | 366 | 54 (d18) | 1600 | 2554 |
| 24 (d37) | 320 | 456 | 55 (d16) | 1600 | 2832 |
| 25 (NA) | 320 | 474 | 56 (d15) | 3200 | 3228 |
| 26 (d47) | 320 | 525 | 57 (d31) | 3200 | 4257 |
| 27 (d12) | 640 | 617 | 58 (NA) | 3200 | 5152 |
| 28 (d9) | 320 | 649 | 59 (d8) | 3200 | 5662 |
| 29 (d10) | 640 | 681 | 60 (NA) | 3200 | 5711 |
| 30 (d27) | 320 | 721 | 61–120 | <20 | <20 |
| 31 (d9) | 640 | 727 | | | |

[a]A total of 120 patient sera were analyzed, including 60 specimens from RT-PCR-confirmed patients (specimens 1–60) and 60 negative specimens (specimens 61–120) that were collected before COVID-19 pandemic (prepandemic).
[b]Sample collection days post after RT-PCR positive test are indicated in parentheses. For some COVID-19-positive specimens, the sample collection days post after RT-PCR positive test are not available (NA).
[c]The NT$_{50}$ and PRNT$_{50}$ values were derived from the reporter virus assay and conventional PRNT assay, respectively.

### Table 2 Cross reactivity of mNG SARS-CoV-2 neutralization assay.

| [a]Immune sera and [b]interfering substances | Sample number | Number of mNG tested positive |
|---|---|---|
| Adenovirus | 1 | 0 |
| Anti-Chikungunya virus | 4 | 0 |
| *Cryptococcus neoformans* antigen | 2 | 0 |
| Anti-Cytomegalovirus | 8 | 0 |
| Anti-Dengue virus | 5 | 0 |
| Anti-Epstein Barr Virus: capsid or nuclear antigen | 8 | 0 |
| Anti-Hepatitis A virus | 5 | 0 |
| Anti-Hepatitis B virus: surface antigen | 15 | 0 |
| Anti-Hepatitis C virus | 3 | 0 |
| Anti-Herpes simplex virus 1 | 7 | 0 |
| Anti-Herpes simplex virus 2 | 5 | 0 |
| Human coronavirus 229E | 1 | 0 |
| Human coronavirus HKU1 | 5 | 0 |
| Human coronavirus NL63 | 1 | 0 |
| Human coronavirus OC43 | 4 | 0 |
| Anti-Human immunodeficiency virus 1 | 10 | 0 |
| Human rhinovirus | 3 | 0 |
| Influenza B virus | 2 | 0 |
| Anti-Measles virus | 7 | 0 |
| Anti-Mumps virus | 5 | 0 |
| Parainfluenza virus 2 | 1 | 0 |
| Parainfluenza virus 4 | 1 | 0 |
| Anti-Parvovirus B19 | 4 | 0 |
| Respiratory syncitial virus | 1 | 0 |
| Anti-Rubella virus | 12 | 0 |
| Anti-Syphilis | 5 | 0 |
| Anti-Toxoplasma | 2 | 0 |
| Anti-Typhus Fever | 1 | 0 |
| Anti-Varicella zoster virus | 13 | 0 |
| Anti-West Nile Virus | 3 | 0 |
| Anti-Yellow fever virus: vaccination | 2 | 0 |
| Anti-Zika virus | 4 | 0 |
| [b]Albumin (4.5 g/dl) | 3 | 0 |
| [b]Elevated bilirubin conjugated (>0.4 mg/dl) | 3 | 0 |
| [b]Elevated bilirubin unconjugated (>0.8 mg/dl) | 3 | 0 |
| [b]Elevated cholesterol (>200 mg/dl) | 3 | 0 |
| [b]Elevated rheumatoid factor (>100 IU/ml) | 3 | 0 |
| [b]Anti-nuclear antibodies | 4 | 0 |

[a]A total of 150 sera with antigens or antibodies against different infections (or immunizations) were tested against mNG SARS-CoV-2 neutralization assay. The immune sera are listed in alphabetical order. Samples tested positive for antibodies against specific pathogens are indicated with prefix "anti", whereas samples tested positive on antigens or pathogen nucleic acids are not indicated with prefix. For the latter group, the specimens were collected within 1 to 6 months after the antigen or PCR tested positive.
[b]A total of 19 samples tested for interfering substances and autoimmune disease nuclear antibodies.

identification of high-neutralizing convalescent plasma for therapy. Treatment of severe COVID-19 patients with convalescent plasma shows clinical benefits[15]. For vaccine development, a standardized neutralizing assay will facilitate down selection of various candidates for clinical development. Furthermore, the reporter assay could be used over time to monitor the waning of protective neutralizing titers in COVID-19 patients and vaccinated individuals, to study the correlates of protection from SARS-CoV-2, and to monitor high-risk populations (such as healthcare workers) for infection prevention. Thus, the ability to rapidly measure neutralizing antibody levels in populations is essential for guiding policymakers to reopen the economy and society, deploy healthcare workers, and prepare for SARS-CoV-2 reemergence.

## Methods

**Cells.** Vero (ATCC®CCL-81) and Vero E6 (ATCC® CRL-1586) were purchased from the American Type Culture Collection (ATCC, Bethesda, MD), and maintained in a high-glucose Dulbecco's modified Eagle's medium (DMEM) supplemented with 10% fetal bovine serum (FBS; HyClone Laboratories, South Logan, UT) and 1% penicillin/streptomycin at 37°C with 5% $CO_2$. All culture medium and antibiotics were purchased from ThermoFisher Scientific (Waltham, MA). All cell lines were tested negative for mycoplasma.

**mNG SARS-CoV-2.** The virus stock of mNG SARS-CoV-2 was produced using an infectious complementary DNA (cDNA) clone of SARS-CoV-2 in which the ORF7 of the viral genome was replaced with reporter mNG gene[10]. After rescued from the genome-length viral RNA-electroporated cells, the viral stock was prepared by amplifying the mNG SARS-CoV-2 on Vero E6 cells for one or two rounds. The titer of the virus stock was determined by a standard plaque assay.

**Human sera and interfering substances.** The research protocol regarding the use of human serum specimens was reviewed and approved by the University of Texas Medical Branch (UTMB) Institutional Review Board. The approved IRB protocol

number is 20-0070. All human serum specimens were obtained at the UTMB. All specimens were de-identified from patient information. A total of 60 de-identified convalescent sera from COVID-19 patients (confirmed with viral RT-PCR positive) were tested in this study. Sixty non-COVID-19 sera, collected before COVID-19 emergence[16,17], were also tested in the reporter virus and PRNT assays. For testing cross reactivity, a total of 150 de-identified specimens from patients with antigens or antibodies against different viruses, bacteria, and parasites were tested in the mNG SARS-COV-2 neutralization assay (Table 2). For testing interfering substances, 19 de-identified serum specimens with albumin, elevated bilirubin, cholesterol, rheumatoid factor, and autoimmune nuclear antibodies were tested in the reporter neutralization assay. All human sera were heat-inactivated at 56°C for 30 min before testing.

**mNG SARS-CoV-2 reporter neutralization assay**. Vero CCL-81 cells ($1.2 \times 10^4$) in 50 μl of DMEM (Gibco) containing 2% FBS (Hyclone) and 100 U/ml Penicillium–Streptomycin (P/S; Gibco) were seeded in each well of black μCLEAR flat-bottom 96-well plate (Greiner Bio-one™). Vero CCL-81 cells, not Vero E6 cells, were selected for the mNG SARS-COV-2 assay to facilitate accurate quantification of fluorescent cells by high-content imaging. The cells were incubated overnight at 37°C with 5% $CO_2$. On the following day, each serum was twofold serially diluted in 2% FBS and 100 U/ml P/S DMEM, and incubated with mNG SARS-CoV-2 at 37°C for 1 h. The virus-serum mixture was transferred to the Vero CCL-81 cell plate with the final multiplicity of infection (MOI) of 0.5. For each serum, the starting dilution was 1/20 with nine two-fold dilutions to the final dilution of 1/5120. After incubating the infected cells at 37°C for 16 h, 25 μl of Hoechst 33342 Solution (400-fold diluted in Hank's Balanced Salt Solution; Gibco) were added to each well to stain cell nucleus. The plate was sealed with Breath-Easy sealing membrane (Diversified Biotech), incubated at 37°C for 20 min, and quantified for mNG fluorescence on Cytation™ 7 (BioTek). The raw images ($2 \times 2$ montage) were acquired using 4× objective, processed, and stitched using the default setting. The total cells (indicated by nucleus staining) and mNG-positive cells were quantified for each well. Infection rates were determined by dividing the mNG-positive cell number to total cell number. Relative infection rates were obtained by normalizing the infection rates of serum-treated groups to those of non-serum-treated controls. The curves of the relative infection rates versus the serum dilutions (log10 values) were plotted using Prism 8 (GraphPad). A nonlinear regression method was used to determine the dilution fold that neutralized 50% of mNG fluorescence ($NT_{50}$). Each serum was tested in duplicates. All mNG SARS-CoV-2 reporter neutralization assay was performed at the BSL-3 facility at UTMB.

**Plaque reduction neutralization test (PRNT)**. Vero E6 cells ($1.2 \times 10^6$ per well) were seeded to six-well plates. On the following day, 100 PFU of infectious clone-derived wild-type SARS-CoV-2 was incubated with serially diluted serum (total volume of 200 μl) at 37°C for 1 h. The virus-serum mixture was added to the pre-seeded Vero E6 cells. After 1 h 37°C incubation, 2 ml of 2% high gel temperature agar (SeaKem) in DMEM containing 5% FBS and 1% P/S was added to the infected cells. After 2 days of incubation, 2 ml neutral red (1 g/l in PBS; Sigma) was added to the agar-covered cells. After another 5-h incubation, neutral red was removed. Plaques were counted for $NT_{50}$ calculation. Each serum was tested in duplicates. The PRNT assay was performed at the BSL-3 facility at UTMB.

**Statistical analysis**. The correlation of the $NT_{50}$ values from mNG reporter SARS-CoV-2 assay and the $PRNT_{50}$ values from plaque neutralization assay was analyzed using a linear regression model in the software Prism 8 (GraphPad). Pearson correlation coefficient and two-tailed p-value are calculated using the default settings in the software Prism 8.

**Reporting summary**. Further information on research design is available in the Nature Research Reporting Summary linked to this article.

## Data availability

Source data are provided with this paper. The assay protocol and sequence of mNG reporter SARS-CoV-2 are provided in the Supplementary Information. The mNG reporter SARS-CoV-2 has been deposited to the World Reference Center for Emerging Viruses and Arboviruses (https://www.utmb.edu/wrceva) at UTMB for distribution.

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

## Acknowledgements
We thank colleagues at UTMB for helpful discussion during the course of this project. P.-Y.S. was supported by NIH grants AI134907, AI145617, and UL1TR001439, and awards from the Sealy & Smith Foundation, Kleberg Foundation, John S. Dunn Foundation, Amon G. Carter Foundation, Gilson Longenbaugh Foundation, and Summerfield Robert Foundation. M.A.G.-B. was supported by NIH grant CA204806 and the Vacek Distinguished Chair. V.D.M. was supported by NIH grants U19AI100625, R00AG049092, R24AI120942, and STARs Award from the University of Texas System. A.E.M. is supported by a Clinical and Translational Science Award NRSA (TL1) Training Core (TL1TR001440) from NIH. C. R.F.-G. is supported by the predoctoral fellowship from the McLaughlin Fellowship Endowment at UTMB.

## Author contributions
P.R., M.A.G.-B., V.D.M., X.X., and P.-Y.S. conceived the study. A.E.M. and C.R.F.-G. performed the experiments and analyzed the results. P.R. prepared the serum specimens. M.A.G.-B., V.D.M., X.X., and P.-Y.S. wrote the manuscript.

## Competing interests
X.X., V.D.M., and P.-Y.S. have filed a patent on the reverse genetic system and reporter SARS-CoV-2. Other authors declare no competing interests.
