## [Peer Review File · Nature Communications]

Reviewers' Comments:

Reviewer #1:

Remarks to the Author:

The manuscript describes a neutralization assay based on a modified SARS-CoV-2 virus that expresses mNeonGreen. The assay enables higher throughput testing of samples compared to plaque reduction assays. A potential benefit of the assay is that it is based on the wildtype virus which could adequately more adequately represent neutralization activity compared to pseudotyped viruses. The associated downside is the further reliance on BSL-3 containment which limits the number of laboratories who may be able to use this assay. The authors showed that the assay results correlated with plaque reduction titers and also confirmed that no unspecific neutralizing activity was observed with sera raised against other infectious agents.

Specific points:

- Since the manuscript is mainly a description of a novel method, an SOP or at least a detailed protocol should be included as supplementary information. It seems like the main purpose of publishing this information would be to enable other laboratories to perform the assay exactly as described.
- While the virus has been deposited, it should be similarly described in detail. A figure showing the exact insertion site and the (codon optimized?) sequence of the construct should be also included in the manuscript.
- Neutralization curves for all samples (including negatives) should be included in addition to the table at least as supplementary data.
- The nomenclature in table 1 is inconsistent, unless the authors in fact spiked in human coronaviruses instead of antisera (which would need to be explained in more detail). Further, "Cross reactivity" should be "Positive tests" or similar.

Minor:

- Line 71: Should be "ORF7"

Reviewer #2:

Remarks to the Author:

In this study, the authors report the use of a patented system to assess sera neutralization at a higher throughput than PRNT in BSL3 conditions. While interesting, the manuscript suffers from some flaws.

Major Comments:

1. The authors report an almost perfect correlation on the analysis of neutralization titers between the PRNT and the assay. However, there was no information on the type of test (parametric or non-parametric).
2. It is unclear how the system bridges the gap for rapid and large-scale serodiagnostics and vaccine evaluation, since it still requires state-of-the-art BSL-3 facilities equipped with high content imaging, which is not readily available in most laboratories.
3. Authors suggest that the system is better than spike pseudotyped lentiviruses or pseudotyped VSV because "One weakness of the spike pseudotyped assay is that it lacks the same composition of an actual virion, including the SARS-CoV-2 M or E proteins.". This argument is incorrect as there is limited literature on anti-M or anti-E neutralizing antibodies. Actually, for the case of SARS, while E

and M can induce antibody responses, the antibodies are not neutralizing (PMC7110836). Moreover, they authors wrongly interpreted the results from the referenced article in line 116 to claim that spike glycoproteins from pseudotyped lentivirus may be different from those on the original SARS-CoV-2 virus.

4. If the authors want to demonstrate value of their system, a direct comparison with the pseudotyped system (BSL2 approved) should be performed.

Minor comments:

5. Methods should be expanded with more details to allow for reproducibility studies.

6. Typo error on line 200.

Reviewer #3:

Remarks to the Author:

In this study the authors aim to determine functional antibodies against SARS CoV-2. They have therefore designed an assay which can be used as an alternative for the classical PRNT, by using a reporter virus mNeonGreen SARS-CoV-2. The assay seems to perform well and has a good correlation to the classical gold standard PRNT. The authors present their study as a rapid high throughput platform, which is true, but it should be mentioned that it still requires a BSL-3 laboratory, making it not very accessible for the majority of laboratories in charge of antibody testing. The paper is therefore mostly a methodological paper describing a novel assay.

Comments :

Line 63 : Although PRNT and ELISA results generally correlate with each other, the lack of complete fidelity of ELISA continues to make PRNT the gold-standard for determining immune protection. The authors should at least discuss the more recent papers in which ELISA correlation to PRNT is being discussed (GeurtsvanKessel et al. medRxiv 2020.04.23.20077156; doi: <https://doi.org/10.1101/2020.04.23.20077156>) , Harvala et al (medRxiv 2020.05.20.20091694; doi: <https://doi.org/10.1101/2020.05.20.20091694>), as well as the availability of a competitive ELISA by Genscript, which claims to correlate to neutralization (large scale validations not yet available). In addition there are pseudoparticle assays available which are not mentioned in this report. Although the assay described in this paper, using a reporter virus is a good alternative for classical PRNT, it is not the most accessible and it is not very likely that the assay will be widely implemented.

Line 65 :However, due to its low throughput, PRNT is not practical for large scale serodiagnosis and vaccine evaluation. This is a major gap for COVID-19 surveillance and vaccine development. The question is, if for serosurveillance purposes it is essential to measure neutralizing antibodies. I think there are better examples in which neutralizing antibodies should be assessed (high risk populations, in infection prevention etc.)

The authors should add more sera from non-SARS CoV-2 coronavirus infected patients to assess specificity.

Reviewer #1 (Remarks to the Author):

The manuscript describes a neutralization assay based on a modified SARS-CoV-2 virus that expresses mNeonGreen. The assay enables higher throughput testing of samples compared to plaque reduction assays. A potential benefit of the assay is that it is based on the wildtype virus which could adequately more adequately represent neutralization activity compared to pseudotyped viruses. The associated downside is the further reliance on BSL-3 containment which limits the number of laboratories who may be able to use this assay. The authors showed that the assay results correlated with plaque reduction titers and also confirmed that no unspecific neutralizing activity was observed with sera raised against other infectious agents.

Response: We thank the reviewer for the constructive suggestions.

Specific points:

- Since the manuscript is mainly a description of a novel method, an SOP or at least a detailed protocol should be included as supplementary information. It seems like the main purpose of publishing this information would be to enable other laboratories to perform the assay exactly as described.

Response: Done. A detailed protocol has been included in Supplementary Methods.

- While the virus has been deposited, it should be similarly described in detail. A figure showing the exact insertion site and the (codon optimized?) sequence of the construct should be also included in the manuscript.

Response: Done. We have indicated the contact information for obtaining the mNeonGreen virus in Data Availability. The mNeonGreen virus has already been shared with CDC, New York State Health Department, many medical centers, and research groups around the world for patient diagnosis and COVID-19 research. In addition, we have added Fig. 1a to depict the exact insertion site and the junction sequences of the mNeonGreen gene. We also provided the genome sequence of the reporter virus in Supplementary Fig. 1.

- Neutralization curves for all samples (including negatives) should be included in addition to the table at least as supplementary data.

Response: Done. We have presented all the neutralization curves in Supplementary Fig. 2.

- The nomenclature in table 1 is inconsistent, unless the authors in fact spiked in human coronaviruses instead of antisera (which would need to be explained in more detail). Further, "Cross reactivity" should be "Positive tests" or similar.

Response: Corrected. We have modified Table 1 as suggested.

Minor:

- Line 71: Should be "ORF7"

Response: Corrected.

Reviewer #2 (Remarks to the Author):

In this study, the authors report the use of a patented system to assess sera neutralization at a higher throughput than PRNT in BSL3 conditions. While interesting, the manuscript suffers from some flaws.

Response: We thank the reviewer for the insightful suggestions.

Major Comments:

1. The authors report an almost perfect correlation on the analysis of neutralization titers between the PRNT and the assay. However, there was no information on the type of test (parametric or non-parametric).

Response: We have added the test type in figure legend and Methods.

2. It is unclear how the system bridges the gap for rapid and large-scale serodiagnostics and vaccine evaluation, since it still requires state-of-the-art BSL-3 facilities equipped with high content imaging, which is not readily available in most laboratories.

Response: The requirement for BSL-3 facility was clearly acknowledged and discussed in the manuscript. Despite the BSL-3 limitation, this assay has a clear utility for high-throughput vaccine evaluation. Indeed, the assay has been used to support COVID-19 vaccine and antiviral development for leading pharmaceutical companies.

3. Authors suggest that the system is better than spike pseudotyped lentiviruses or pseudotyped VSV because “One weakness of the spike pseudotyped assay is that it lacks the same composition of an actual virion, including the SARS-CoV-2 M or E proteins.”. This argument is incorrect as there is limited literature on anti-M or anti-E neutralizing antibodies. Actually, for the case of SARS, while E and M can induce antibody responses, the antibodies are not neutralizing (PMC7110836). Moreover, they authors wrongly interpreted the results from the referenced article in line 116 to claim that spike glycoproteins from pseudotyped lentivirus may be different from those on the original SARS-CoV-2 virus.

Response: Corrected. We have now removed the questionable statement. Please see more elaborations in response to Comment 4.

4. If the authors want to demonstrate value of their system, a direct comparison with the pseudotyped system (BSL2 approved) should be performed.

Response: We thank the reviewer for this important point. Respectfully, we think the requested studies are beyond the scope of the current manuscript. This is because the requested task represents a completely new project that requires a series of experiment to establish a different assay platform, including (i) construction of the pseudotyped virus; (ii) optimization of the

production of pseudotyped virus; (iii) evaluation of the pseudotyped virus for neutralization; and (iv) testing clinical specimens. We currently do not have the pseudotyped virus system in hand. In addition, the volumes of patient sera were very limited and many of them have already been used up. To address reviewer's point, we have deleted the speculative statement about the pseudotyped VSV assay platform (see response to Comment 3). In addition, to strengthen the current results, we have added results from 70 more human specimens (20 more COVID-19-positive sera and 50 more COVID-19-negative specimens) in the revised manuscript.

Minor comments:

5. Methods should be expanded with more details to allow for reproducibility studies.

Response: Done. A detailed protocol has been included in Supplementary Methods.

6. Typo error on line 200.

Response: Corrected.

Reviewer #3 (Remarks to the Author):

In this study the authors aim to determine functional antibodies against SARS CoV-2. They have therefore designed an assay which can be used as an alternative for the classical PRNT, by using a reporter virus mNeonGreen SARS-CoV-2. The assay seems to perform well and has a good correlation to the classical gold standard PRNT. The authors present their study as a rapid high throughput platform, which is true, but it should be mentioned that it still requires a BSL-3 laboratory, making it not very accessible for the majority of laboratories in charge of antibody testing. The paper is therefore mostly a methodological paper describing a novel assay.

Response: We thank the reviewer for the positive and constructive comments.

Comments:

Line 63 : Although PRNT and ELISA results generally correlate with each other, the lack of complete fidelity of ELISA continues to make PRNT the gold-standard for determining immune protection. The authors should at least discuss the more recent papers in which ELISA correlation to PRNT is being discussed (GeurtsvanKessel et al. medRxiv 2020.04.23.20077156; doi: <https://doi.org/10.1101/2020.04.23.20077156>) , Harvala et al (medRxiv 2020.05.20.20091694; doi: <https://doi.org/10.1101/2020.05.20.20091694>), as well as the availability of a competitive ELISA by Genscript, which claims to correlate to neutralization (large scale validations not yet available). In addition there are pseudoparticle assays available which are not mentioned in this report.

Response: We thank the reviewer for this important suggestion. We have included and discussed these ELISA and pseudoparticle assays.

Although the assay described in this paper, using a reporter virus is a good alternative for classical PRNT, it is not the most accessible and it is not very likely that the assay will be widely implemented.

Response: We agree with the reviewer on the BSL-3 limitation. We have clearly acknowledged and discussed this weakness in the original manuscript. Despite this limitation, this assay has a

clear utility for high-throughput vaccine evaluation. In fact, the assay has been used to support COVID-19 vaccine and antiviral development for leading pharmaceutical companies. Meanwhile, we are engineering an attenuated version of SARS-CoV-2 so that the assay could be performed at a BSL-2 facility.

Line 65 :However, due to its low throughput, PRNT is not practical for large scale serodiagnosis and vaccine evaluation. This is a major gap for COVID-19 surveillance and vaccine development. The question is, if for serosurveillance purposes it is essential to measure neutralizing antibodies. I think there are better examples in which neutralizing antibodies should be assessed (high risk populations, in infection prevention etc.)

Response: We thank the reviewer for bringing up this point. We have now included the suggested target individuals (such as healthcare workers) who should be prioritized to be tested for their neutralizing antibody titers.

The authors should add more sera from non-SARS CoV-2 coronavirus infected patients to assess specificity.

Response: We have added 70 more serum specimens (20 more COVID-19-positive patients and 50 more COVID-19-negative patients) to the study. The results have been added to Fig. 1e. In addition, we have added 12 more non-COVID-19 patient specimens to strengthen the specificity testing results in Table 1. However, the specimen number for non-SARS-CoV-2 coronaviruses remains low. Nevertheless, the increased total number of patient specimens has allowed us to further demonstrate the assay specificity.